# The Vasoactive Effect of Perivascular Adipose Tissue and Hydrogen Sulfide in Thoracic Aortas of Normotensive and Spontaneously Hypertensive Rats

**DOI:** 10.3390/biom12030457

**Published:** 2022-03-16

**Authors:** Samuel Golas, Andrea Berenyiova, Miroslava Majzunova, Magdalena Drobna, Muobarak J. Tuorkey, Sona Cacanyiova

**Affiliations:** 1Center of Experimental Medicine, Institute of Normal and Pathological Physiology, Slovak Academy of Sciences, 84104 Bratislava, Slovakia; samuel.golas@savba.sk (S.G.); andrea.berenyiova@savba.sk (A.B.); miroslava.majzunova@savba.sk (M.M.); magdalena.drobna@savba.sk (M.D.); 2Department of Animal Physiology and Ethology, Faculty of Natural Sciences, Comenius University, 81108 Bratislava, Slovakia; 3Zoology Department, Division of Physiology, Faculty of Science, Damanhour University, Damanhour 22514, Egypt; physio_mj_tuorkey@yahoo.com

**Keywords:** adipose tissue, H_2_S, thoracic aorta, Wistar, SHR, essential hypertension

## Abstract

The objective of this study was to investigate the vasoregulatory role of perivascular adipose tissue (PVAT) and its mutual interaction with endogenous and exogenous H_2_S in the thoracic aorta (TA) of adult normotensive Wistar rats and spontaneously hypertensive rats (SHRs). In SHRs, hypertension was associated with cardiac hypertrophy and increased contractility. Regardless of the strain, PVAT revealed an anticontractile effect; however, PVAT worsened endothelial-dependent vasorelaxation. Since H_2_S produced by both the vascular wall and PVAT had a pro-contractile effect in SHRs, H_2_S decreased the sensitivity of adrenergic receptors to noradrenaline in Wistar rats. While H_2_S had no contribution to endothelium-dependent relaxation in Wistar rats, in SHRs, H_2_S produced by the vascular wall had a pro-relaxant effect. We observed a larger vasorelaxation induced by exogenous H_2_S donor in SHRs than in Wistar rats. Additionally, in the presence of PVAT, this effect was potentiated. We demonstrated that PVAT of the TA aggravated endothelial function in SHRs. However, H_2_S produced by the TA vascular wall had a pro-relaxation effect, and PVAT revealed anti-contractile activity mediated by the release of an unknown factor and potentiated the vasorelaxation induced by exogenous H_2_S. All these actions could represent a form of compensatory mechanism to balance impaired vascular tone regulation.

## 1. Introduction

Several studies have confirmed a dual, endocrine and paracrine, function of perivascular adipose tissue (PVAT) and its contribution to vascular tone regulation via the production of vasoactive substances with anticontractile or pro-contractile effects [1,2,3]. PVAT can reveal regional differences and participate in vascular tone regulation in a special way depending on vessel type [4]. Under physiological conditions, PVAT exerts predominantly anticontractile effects, which are induced, in addition to others, by a transferable factor called adipocyte-derived relaxing factor (ADRF) [5]. One of the substances that could be an ADRF is hydrogen sulfide (H_2_S) [6]. However, H_2_S can also be produced by vessel wall–smooth muscle cells and endothelial cells [7,8], thus regulating various processes. Literary data and our previous results also confirmed a biphasic concentration-dependent vasoactive effect of H_2_S, vasoconstrictor and vasorelaxant, in rat conduit arteries [9,10]. Moreover, H_2_S can be involved in vascular tone regulation via interaction with other signaling pathways, and the crosstalk between H_2_S and NO signaling has been confirmed by several authors [11,12]. H_2_S can also stimulate enzymatic and nonenzymatic endogenous antioxidants through several transcription factors, such as Nrf2 and nuclear factor-κB, in different parts of the cardiovascular system [13]. Taken together, endogenous H_2_S can operate in vascular tone control and blood pressure regulation in distinct ways, and its role in various pathological conditions remains unexplained.

Under pathological conditions of essential hypertension, the activity of PVAT and the vasoactive effect of substances produced by PVAT and the vascular wall may be changed [14]. Several papers confirmed that the increased activity of the sympathetic nervous system promoted increased lipolysis, which could result in lost or reduced anticontractile effects of PVAT, thus participating in the development of hypertension [15,16]. On the other hand, Zemancikova and Torok [17] observed that a moderate increase in body adiposity did not potentiate the anticontractile effect of PVAT in SHRs. This suggests that qualitative alterations of various signal pathways, rather than quantitative changes in PVAT, are important when considering its involvement in increased vascular tone in SHRs. The primary site of the initial inflammation in hypertension is within the PVAT and PVAT/adventitial border and PVAT-derived free radicals can cause oxidation of LDL-cholesterol, whereas oxidized LDL is a major player in the development of atherogenesis and endothelial dysfunction. Jain et al. [18] demonstrated that a positive relationship exists between the blood levels of H_2_S and HDL-cholesterol versus a negative relationship with the LDL/HDL ratio, indicating that circulating levels of H_2_S could be a regulator of HDL and LDL homeostasis in the blood. The authors suggested that H_2_S suppressed the oxidation of LDL-cholesterol, which prevented endothelial dysfunction and the progression of atherosclerosis. From this point of view, it seems that there could be a link between changes in the plasma lipid profile and the sulfide signaling pathway, especially in relation to PVAT-mediated H_2_S effects.

Several authors have investigated the relationship between PVAT action and the sulfide signaling pathway thus far. In conditions of essential hypertension, decreased H_2_S levels, as well as CSE expression and activity, have been observed [19]. The authors showed that a reduced amount of endogenous H_2_S and an associated, reduced vasorelaxant effect resulted in an increase in vascular tone and acted as an important pathologic factor involved in the development of primary hypertension. In our previous study, we confirmed the pro-contractile effect of H_2_S produced by the mesenteric arterial wall of spontaneously hypertensive rats (SHRs), and although the presence of PVAT eliminated this pro-contractile effect, it was mediated by a factor other than H_2_S [3]. One of the possible mechanisms of H_2_S-induced vasoconstriction could be the interaction with reactive oxygen species. Zong et al. [20] showed that tirone, a mimetic of superoxide dismutase, significantly inhibited NaHS-induced vasoconstriction. It seems that a superoxide radical might mediate H_2_S-mediated vascular pro-contractile responses. In an experimental model of metabolic syndrome and prediabetes, we showed that H_2_S within the arterial wall contributed to endothelial dysfunction; however, PVAT of the thoracic aorta was endowed with compensatory vasoactive mechanisms, which included stronger anticontractile action of H_2_S [21]. Several authors confirmed that exogenous H_2_S could also take part in compensatory vasoactive responses. In SHRs, Liu et al. [9] proved that, after administration of H_2_S, hypertension and arterial remodeling were partially prevented. Berenyiova et al. [22] demonstrated that the increased H_2_S-induced vasorelaxation after acute NO inhibition could be considered a salvage mechanism in cases of endogenous NO deficiency. In the mesenteric artery of SHRs, we observed similar results; moreover, it was confirmed that the presence of PVAT conditioned the potentiation of the vasorelaxant effect of exogenous H_2_S, indicating a close relationship between the vasorelaxant action of H_2_S and PVAT [3].

The above-mentioned results suggest that both PVAT and the sulfide signaling pathway could exhibit dual vasoactive action depending on the type of triggered signaling pathway. This balance between impaired regulation of vascular function and compensatory vasoactive mechanisms seems to be affected by other factors, and the activity of PVAT and interaction with donor H_2_S or its relationships with endogenously produced H_2_S can vary within the vessel type. Torok et al. [23] demonstrated that, unlike in the mesenteric artery, the anticontractile effect of PVAT in the abdominal aorta of SHRs has disappeared. However, data about the relationship among the sulfide signaling pathway, PVAT and its impact on vasoactivity in the thoracic aorta of SHRs are completely missing. The objective of this study was to evaluate and compare the participation of PVAT and endogenously produced H_2_S in the contractile and relaxant responses as well as the effect of exogenous H_2_S in isolated thoracic aortas of normotensive Wistar rats and SHRs. Moreover, an important factor in the development and maintenance of hypertension could be oxidative stress, during which there is an overproduction of oxidants and a reduction in the antioxidant capacity and bioavailability of NO in the vascular wall [24]. Changes in the state of the oxidative status could also be associated with the vasoactive effects of the sulfide pathway; therefore, the redox state in the heart and aorta was determined through the measurement of superoxide levels.

## 2. Materials and Methods

### 2.1. Guide for the Use and Care of Laboratory Animals

Animals were bred in accordance with the institutional guidelines of the State Veterinary and Food Administration of the Slovak Republic and the Committee on the Ethics of Procedures in Animal, Clinical and other Biomedical Experiments (Permit Number: EC/CEM/2017/4) of the Centre of Experimental Medicine, as well as in accordance with the European Convention for the Protection of Vertebrate Animals used for Experimental and other Scientific Purposes, Directive 2010/63/EU of the European Parliament. All rats used in this study were received from an accredited breeding establishment of the Institute of Normal and Pathological Physiology, Centre of Experimental Medicine, Slovak Academy of Sciences, and were housed under a 12 h light–12 h dark cycle at a constant humidity (45–65%) and temperature (20–22 °C) with free access to standard laboratory rat chow and drinking water (ad libitum).

### 2.2. Experimental Animals and Basic Parameters

Sixteen-week-old male Wistar rats (*n* = 8) and SHRs (*n* = 8) were used in this study. Systolic blood pressure (SBP) was measured in prewarmed rats by a noninvasive plethysmography method on rat tail arteries before the beginning of the in vitro study. The body weight (BW) of each rat was determined before decapitation. Rats were killed by decapitation after a brief anesthetization with CO_2_; the heart and retroperitoneal fat were weighed, tibia length was measured, and the thoracic aorta (TA) was isolated. Plasma samples were collected just after decapitation and frozen in aliquots for biochemical determinations. The basic levels of cholesterol (Chol), high-density lipoprotein cholesterol (HDL), low-density lipoprotein cholesterol (LDL), triacylglycerols (TAG), and glucose (GLU) were commercially determined in Laboklin GmbH. The values of parameters represented postprandial concentrations. We avoided 24 h fasting because fasting stimulates lipid mobilization and lipolysis, which could reduce TA PVAT.

### 2.3. Functional Study

To examine the vasoactive properties, the TA was cleaned of connective tissue, cut into 5 mm long rings with intact endothelium, and divided into two groups, without PVAT (PVAT−) and with intact PVAT (PVAT+), to distinguish between the contribution of PVAT and endogenous H_2_S produced by the arterial wall and the effect of total H_2_S produced by the arterial wall and surrounding perivascular fat. Subsequently, the rings were vertically fixed between two stainless wire triangles in a 20 mL incubation organ bath with Krebs solution (in mmol/L: 118 NaCl, 5 KCl, 25 NaHCO_3_, 1.2 MgSO_4_, 1.2 KH_2_PO_4_, 2.5 CaCl_2_, 11 glucose, 0.032 CaNa_2_EDTA). The solution was oxygenated with 95% O_2_ and 5% CO_2_ and kept at 37 °C. The upper triangles were connected to isometric tension sensors (FSG-01, MDE, Budapest, Hungary), and changes in tension were registered by an AD converter NI USB-6221 (MDE, Budapest, Hungary). Changes in isometric tension were registered by SPEL Advanced Kymograph (MDE, Budapest, Hungary) software. A resting tension of 1 g was applied to each ring and maintained throughout a 45 to 60 min equilibration period until stress relaxation no longer occurred.

KCl (125 mmol/L, physiological Krebs solution was changed to a solution in which NaCl was exchanged for an equimolar concentration of KCl) was added to the organ bath for only 2 min to confirm the sufficient contractility of the sample. The presence of functional endothelium was assessed in all preparations by determining the ability of acetylcholine (10^−5^ mol/L) to induce relaxation in noradrenaline (NA) pre-contracted (10^−6^ mol/L) arteries. After washing with physiological Krebs solution and an equilibration period, experiments with NA were started so as to obtain the contractile responses. Adrenergic contractions were determined in the TA as the responses to cumulatively applied exogenous NA (10^−10^–3 × 10^−6^ mol/L). The rings were then exposed to cumulative doses of acetylcholine (Ach; 10^−10^–10^−5^ mol/L) on the NA-precontracted arterial rings.

To evaluate the participation of endogenous H_2_S, contractile and vasorelaxant responses induced by vasoactive substances (Na, Ach) were followed before and 20 min after acute administration of DL-propargylglycine (10 mmol/L), an inhibitor of cystathionine-γ-lyase (CSE).

All contractile responses were expressed in g as developed tension to demonstrate maximum reached contraction. To demonstrate the sensitivity of the arterial wall, all individual curves were expressed as a percentage of the maximum tissue response to NA (not shown), and the concentrations of NA producing the half-maximum response (EC_50_) were calculated and expressed as the negative logarithm of NA molar concentration. The rate of relaxation was expressed as a percentage of the contractile agonist-induced contraction.

Na_2_S was used to evaluate the vasoactive effect of exogenous H_2_S. Na_2_S dissociates in water solution to Na^+^ and S^2−^, which reacts with H^+^ to yield HS^−^ and H_2_S. We use the term “Na_2_S” to encompass the total mixture of H_2_S, HS^−^, and S^2−^. Direct vasoactive effects of Na_2_S were observed on NA-precontracted (10^−6^ mol/L) rings by administration of increasing doses of Na_2_S (20, 40, 80, 100, 200, and 400 μmol/L). The rate of relaxation was expressed as a percentage of the NA-induced contraction.

### 2.4. Measurement of Superoxide Production in Selected Tissues

The level of superoxide anions was measured by a chemiluminescent method using lucigenin based on the intensity of the emitted photons. Heart and vessel samples were placed in Krebs solution immediately after removal. Oxygenated (mixed 95% O_2_, 5% CO_2_) 50 μmol/L lucigenin and samples in oxygenated Krebs solution were incubated in the dark for 20 min at 37 °C. After incubation, either background chemiluminescence or chemiluminescence produced by the used tissues was measured using a TriCarb 2910TR liquid scintillation analyzer (PerkinElmer, Waltham, MA, USA). The resulting mean values were expressed as cpm/mg (count per minute/milligram) of tissue.

### 2.5. Western Blotting

PVAT and aortic samples were homogenized on ice in 0.05 M Tris-HCl buffer (pH 7.4) supplemented with protease inhibitors. The protein concentrations were determined with a BCA assay kit (Millipore Sigma, 71285, Burlington, MA, USA). Proteins (20 μg total protein for PVAT and 5 μg total protein for aorta) were separated by 10% SDS–PAGE and transferred to nitrocellulose membranes. The membranes were blocked with 5% milk in Tris-buffered saline containing Tween 20 (TBS-T). Afterward, the membranes were incubated with a mouse monoclonal anti-CSE antibody (Proteintech^®^, Manchester, UK; dilution 1:2000) for 2 h at room temperature. Following 3 washes (3 × 10 min) with TBS-T, the membranes were incubated with an anti-rabbit IgG HRP-linked antibody (Cell Signaling Technology, MA, USA; dilution 1:2000) for 1 h at room temperature. Both primary and secondary antibodies were diluted in TBS-T containing 1% milk. All blots were reprobed with a mouse monoclonal anti-GAPDH antibody (Santa Cruz Biotechnology, Dallas, TX, USA; dilution 1:2000) for 1 h at room temperature for PVAT samples or with a mouse monoclonal anti-α-actin antibody (Sigma–Aldrich, Saint Louis, MO, USA; dilution 1:1000) and incubated overnight at 4 °C for aorta samples. The signals were visualized with Clarity Western ECL Substrate (Bio–Rad, 1705061, Hercules, CA, USA) using a ChemiDocTM Touch Imaging System (Bio–Rad) and quantified with Image Lab Software. Target protein amounts were normalized to GAPDH for PVAT samples or to α-actin for aorta samples and are presented in arbitrary units (a.u.).

### 2.6. Statistical Analysis

The data are expressed as the mean ± S.E.M. For the statistical evaluation of vasoactive responses between groups, three-way analysis of variance (ANOVA) for repeated measurements with the Bonferroni post hoc test was used to evaluate (a) the effect of strain and PVAT and (b) the effect of PVAT and inhibitor. To evaluate general cardiovascular and plasmatic parameters, the Student’s *t*-test was used. Differences between means were considered significant at *p* < 0.05.

### 2.7. Drugs

The following drugs were used: propargylglycine, acetylcholine, sodium sulfide nonahydrate from Sigma–Aldrich (St Louis, MI, USA), and noradrenaline from Zentiva (Prague, Czech Republic). All drugs were dissolved in distilled water.

## 3. Results

### 3.1. General Characteristics of Experimental Animals

The comparison of the basic cardiovascular and functional parameters between Wistar rats (*n* = 8) and SHRs (*n* = 8) is shown in Table 1. In SHRs, significantly higher values of SBP were observed (*p* < 0.01). SHRs had reduced body weight compared to Wistar rats (*p* < 0.001). However, the heart weight/body weight and heart weight/tibia length ratios were increased in this strain (*p* < 0.001 and *p* < 0.05). There were no differences in the amount of retroperitoneal fat, adiposity of the body, glucose level, or basic lipid profile determined in the plasma of Wistar rats and SHRs, with the exception of TAG, which was significantly decreased in SHRs compared to Wistar rats (*p* < 0.01).

### 3.2. Role of Perivascular Adipose Tissue and Vasoactive Effect of H_2_S

The cumulative application of exogenous NA (10^−10^–10^−5^ mol/L) induced vasoconstriction in a concentration-dependent manner (*n* = 8). The presence of PVAT (F_(1;262)_ = 36.79; *p* = 5.67 × 10^−9^) and the strain (F_(1;262)_ = 52.07; *p* = 8.56 × 10^−12^) significantly affected these adrenergic vasocontractions. SHRs revealed a significantly increased contractile response to exogenous NA in both PVAT− (*p* < 0.001) and PVAT+ (*p* < 0.001) rings compared with normotensive Wistar rats (Figure 1a). PVAT reduced the contractile responses in Wistar rats (*p* < 0.001) and SHRs (*p* < 0.001). In both SHRs and Wistar rats, PVAT had no effect on the sensitivity of adrenergic receptors to exogenous NA. Alternatively, PVAT– (*p* < 0.01) and PVAT+ (*p* < 0.05) rings of SHRs revealed significantly increased sensitivity to NA compared with Wistar rats (Figure 1b, Table 2). The participation of endogenously produced H_2_S in adrenergic contractile responses of the TA was tested after acute pretreatment with PPG (10 mmol/L, 20 min) in PVAT+ and PVAT– rings. The changes in basal tension induced by pretreatment with PPG are shown in Table 3, and they confirmed the prorelaxant effect of basally released H_2_S in PVAT– rings of SHRs. Acute pretreatment with PPG had no impact on adrenergic contraction in the Wistar rats, regardless of PVAT presence (Figure 1c); on the other hand, in the presence of PPG, PVAT rings (*p* < 0.01) of Wistar rats showed increased sensitivity to NA (Table 2). In SHRs, acute pretreatment with PPG decreased the contractile responses in rings with PVAT+ (*p* < 0.05) and PVAT− (*p* < 0.001), whereas the smallest contraction was observed in PVAT− preserved arteries (*p* < 0.001) (Figure 1d). The changes in sensitivity were not observed.

The application of Ach (10^−10^–10^−5^ mol/L) relaxed the NA-precontracted TA rings, and the presence of PVAT significantly inhibited the vasorelaxant response in Wistar rats (*p* < 0.001) and SHRs (*p* < 0.001). However, there was no difference between the vasorelaxation responses of the TA in Wistar rats (*n* = 8) compared to SHR (*n* = 8), and the presence of PVAT did not affect it (Figure 2a). In Wistar rats, pretreatment with PPG did not affect Ach-induced vasorelaxation in rings without PVAT or with preserved PVAT (*n* = 8) (Figure 2b). In the SHRs, pretreatment with PPG did not affect vasorelaxation in PVAT+ aortic rings; however, vasorelaxation of PVAT− aortic rings (*p* < 0.001) was significantly reduced after PPG incubation (Figure 2c).

The application of cumulative concentrations of the H_2_S donor Na_2_S·9H_2_O (20, 40, 80, 100, 200, and 400 μmol/L) to NA-precontracted TA rings induced a biphasic effect in both Wistar rats and SHRs. Lower concentrations of Na_2_S induced contraction (20, 40, and 80 µmol/L), whereas higher concentrations (100, 200, and 400 μmol) evoked vasorelaxation of the arterial wall (Figure 3).

The effect of strain (F_(1;184)_ = 74.94, *p* = 4.66 × 10^−15^) on vasoactive responses induced by exogenous H_2_S was confirmed. In the SHRs, regardless of the presence of PVAT, we confirmed an increased maximal vasorelaxant phase of the Na_2_S-induced response (*p* < 0.001) compared with that in Wistar rats. In Wistar rats, it was observed that PVAT did not affect the biphasic vasoactive effect of H_2_S; however, in SHRs, the rings with PVAT revealed significantly increased vasorelaxation induced by exogenous H_2_S (*p* < 0.05).

### 3.3. Measurement of Superoxide Production in Selected Tissues

A significant increase in the level of superoxide measured in the aortic tissue (*n* = 8; *p* < 0.05) and heart left ventricle (*n* = 8, *p* < 0.01) was observed in SHRs compared to Wistar rats (Figure 4).

### 3.4. Protein Expression of Cystathionine γ-Lyase

The protein expression of CSE was comparable between Wistar rats and SHRs in both tissues—thoracic aorta and PVAT (Figure 5).

## 4. Discussion

Mutual interaction between PVAT and H_2_S modulates vascular tone under normotensive conditions. Under hypertensive conditions, the activity of PVAT and vasoactive effects of H_2_S may be changed. Our findings showed that, in addition to pathological changes developed in essential hypertension, compensatory mechanisms, including the anticontractile action of PVAT, potentiation of the vasorelaxant effect to exogenous H_2_S, and participation of arterial-wall H_2_S in maintaining endothelial function, could be triggered to balance vascular tone.

### 4.1. PVAT-Related Modulation of Vasoactive Responses

In this study, we demonstrated that contractile responses of the TA to exogenous noradrenaline were significantly increased in SHRs; moreover, regardless of PVAT, we confirmed the increased sensitivity to exogenous noradrenaline compared to Wistar rats. However, the presence of PVAT reduced the contractile response of the TA, not only in normotensive rats but also in hypertensive rats. Similar results were found in our previous experiments on the mesenteric artery of SHRs [3], but also in the TA of hypertriglyceridemic rats (HTGs), a model of metabolic syndrome [21]. In the present study, we found a reduced level of triacylglycerides and an unchanged amount of visceral fat, while in the mentioned model of HTG rats, increased values for both parameters were recorded; therefore, it seems that changes in body adiposity may not be directly associated with PVAT-related anticontractile properties. However, the resulting anticontractile effect of PVAT can depend on the types of studied arteries. Torok et al. [23] confirmed that, although PVAT inhibited noradrenaline-induced contractions in mesenteric arterial rings, no change due to PVAT presence was seen in the abdominal aorta of SHRs. Padilla et al. [25] confirmed that thoracic and abdominal PVATs display striking differences in their phenotype; PVAT surrounding TAs was composed of multilocular brown adipocytes and abundant mitochondria, whereas adipocytes surrounding abdominal aortas were primarily unilocular, thus resembling white adipose tissue. It seems that phenotypic differences in PVAT depots could contribute to the heterogeneous modulation of underlying vascular cells. Although both brown and white adipocytes have a secretory function, brown adipocytes produce significantly lower numbers of hormones and cytokines, including resistin and angiotensinogen, which are abundantly produced in white adipocytes [26]. An important finding of Lu et al. [14] was the structural change in the increased number of brown PVAT adipocytes in the TAs of SHRs compared to WKY rats. The authors suggested that an increased proportion of brown adipocytes may lead to lesser production of angiotensin (1–7), at least partially explaining the PVAT dysfunction observed in the TA of SHRs. Our findings confirming the preserved anticontractile properties of PVAT in this vessel contrast with their findings; however, this discrepancy can be explained using a different control group, contractile agonist, or pretension value. Indeed, it seems that the value of pretension could affect the anticontractile effect of PVAT. In our previous study, we used a pretension of 0.7 g when PVAT of the mesenteric artery had a strong anticontractile effect in SHRs. On the other hand, Torok et al. [23] used pretension 1 g in the same artery and strain and confirmed the weaker anticontractile effect of PVAT. In this study, we used a pretension of 1 g, while Lu et al. [14] used a computerized myograph system to set 3 g of preload in the TA. The use of different preloads could be a reason for the difference in the anticontractile effect of PVAT, which was preserved in our experiment but damaged in their experiment.

In the next part of our study, we investigated acetylcholine-induced vasorelaxation as a marker of endothelial function. Regardless of strain, PVAT significantly reduced acetylcholine-induced vasorelaxation. However, we did not observe a difference in acetylcholine-induced vasorelaxation between SHRs and Wistar rats. Despite the described endothelial dysfunction responsible for the development of hypertension in SHRs, in our study, endothelial dysfunction in the TA was absent in 16-week-old SHRs. Our results are partially consistent with those of our previous study, which showed that the TA of adult SHRs revealed comparable endothelium-dependent relaxation and preserved the NO signaling pathway with respect to normotensive rats; nevertheless, the maximal response was significantly reduced [22]. Tang et al. [27] reported that, in adult SHRs, during the developed phase of hypertension, the effectiveness of NO can be decreased through the overproduction of superoxides. In our study, we found elevated concentrations of superoxides in the heart and aorta of SHRs; however, it did not affect endothelium-dependent relaxation. We suggested that alternative and/or compensatory vasoactive mechanisms, such as the production of NO from nitrites and nitrates, or the neuronal isoform of NO synthase previously confirmed in the arteries of SHRs [28,29], could be activated. The inhibitory effect of PVAT on acetylcholine-induced vasorelaxation observed in this study agrees with the previous findings of several authors [19,21,30,31,32]. Payne et al. [31] suggested impaired endothelial NO production of periadventitial adipose tissue-derived factors via site-specific inhibition of NO-synthase phosphorylation in canine coronary arteries in normotensive conditions. Another study reported that the deteriorative effect of PVAT was associated with diminished NO production, whereas the protein expression and activity of endothelial NO synthase were not significantly affected [32]. In PVAT of SHRs, inhibited activity of NO synthase was noticed; however, it was accompanied by increased endothelial NO synthase expression [19,21,30]. Moreover, the adipose tissue of SHR arteries is characterized by increased oxidative stress [33], and Melrose [34] showed that increasing free radical production can reduce the vasorelaxant effect of NO. Additionally, summarizing all data, PVAT of hypertensive aortas worsened endothelial function, while it triggered anti-contractile activity, probably as a compensatory mechanism.

### 4.2. PVAT—Hydrogen Sulfide Interaction

Several authors have investigated the role of H_2_S in various models of hypertension. Previous data suggested that endogenous H_2_S could contribute to the etiopathogenesis of essential hypertension since decreased H_2_S production has been observed in the plasma of SHRs [19,30]. Under such pathological conditions, the balance between lipid metabolism (which includes fat storage in the arterial wall), H_2_S signaling pathways, and vasoactive regulatory mechanisms may be impaired. It appears that there could be a close relationship between H_2_S signaling and PVAT activity. However, the role of PVAT-derived H_2_S is species-specific and may also vary within a vessel type [35,36].

Al-Magableh and Hart [37] demonstrated that the CSE-H_2_S signaling pathway regulates basal vascular tone, as administration of CSE inhibitors induced contraction of the vascular wall. In our study, we also confirmed that the addition of PPG induced a mild contraction in PVAT-denuded rings in SHRs, compared to the control group. This suggests that, in the regulation of basal tone, H_2_S released from the vascular wall but not from PVAT had a pro-relaxant effect, contributing to vascular tone reduction. However, in the mesenteric artery, in contrast to the TA, we previously confirmed the pro-contractile effect of H_2_S released from the arterial wall [3]. In conditions of essential hypertension, hyperactivity of the sympathetic nervous system has been confirmed: centrally increased basal tone and decreased threshold potential for contractile responses [38], which could facilitate the manifestation of pro-contractile action of H_2_S in the mesenteric artery. However, it seems that the lower density of sympathetic innervation and decreased sympathetic activity in the TA compared to the mesenteric artery [39] might explain the preserved pro-relaxant effect of H_2_S basally produced by the arterial wall. Regardless, the pathological background and the increased sympathetic draft of SHRs go hand in hand with the finding that the active increase in vascular tone disabled the pro-relaxant effect of H_2_S basally produced by the aortic wall.

Fang et al. [40] showed that H_2_S produced by PVAT could have an anticontractile effect in conditions of pressure-overloaded hypertension induced by abdominal aortic banding. The authors showed that transplanting PVAT into the periadventitia of stenotic aortas ameliorated the elevated arterial blood pressure and decreased the angiotensin II level in the aorta. The authors hypothesized that aortic PVAT began to produce H_2_S, which reduced vascular tone, as a compensatory response to hypertension. In this study, pretreatment with PPG revealed a pro-contractile effect of H_2_S in adrenergic contractile responses regardless of the presence of PVAT. In the mesenteric arteries of SHRs, we observed that H_2_S produced by arterial tissue, but not PVAT, contributed to an adrenergic vasoconstrictor response by a pro-contractile effect, which predominantly issued from the ability to increase the sensitivity of the arterial wall to noradrenaline [3]. Contemporarily, PVAT of both the TA (this study) and mesenteric artery (previous study) retained the ability to reduce adrenergic contractile responses, which was not affected by PPG pretreatment. These findings show that PVAT of arteries in SHRs was able to produce some unknown anticontractile factors, other than H_2_S, as a compensatory mechanism to balance increased vascular tone. However, in a nonobese model of metabolic syndrome, HTG rats, we confirmed that H_2_S produced from PVAT of the TA had an anticontractile effect that was associated with the significantly increased expression of CSE and H_2_S vasoactive action [21]. This result suggested that metabolic disorders associated with only a mild increase in blood pressure did not affect the ability of PVAT to release H_2_S and trigger vasoactive compensatory mechanisms. Under conditions of essential hypertension, the expression of the H_2_S-producing enzyme in both the PVAT and the vascular wall of the TA was comparable to that in the control group, and although we cannot rule out the changes in enzyme activity and the regulation of H_2_S levels by other enzymes (e.g., CBS), the development of genetically determined hypertension was associated with the exclusion of H_2_S involvement in the PVAT-related compensatory vasoactive effect.

We also focused on the role of endogenous H_2_S in endothelium-dependent vasorelaxation. Several studies have shown that H_2_S produced from the vascular wall of the aorta could be involved in vasorelaxation due to its interaction with NO and contribution to its vasorelaxant effect [41,42]. However, we did not observe any effect of H_2_S produced by the vascular wall or PVAT on the Ach-induced endothelium-dependent vasorelaxant response of the TA in Wistar rats. In SHRs, we confirmed the pro-relaxant effect of H_2_S produced by the vascular wall, which may represent a compensatory mechanism to offset impaired vascular tone. Additionally, pretreatment with PPG revealed that H_2_S produced by PVAT did not have any effect on vasorelaxation. Most likely, there are some unknown factors released by PVAT that diminish the pro-relaxant action of H_2_S produced by the arterial wall. This agrees with the evidence of several authors who confirmed that an unknown factor (or factors) released from PVAT may block the vasoactive effects of substances released from the endothelium under hypertensive conditions. Lee et al. [43] confirmed a reduced vasoconstriction after treatment with AII receptor antagonist losartan, indicating an increased release of angiotensin II in PVAT of the TA in SHRs compared to Wistar Kyoto rats. In addition, they observed a reduced release of palmitic acid methyl ester from PVAT, which is a potent vasodilator opening Kv channels. Angiotensin II may inhibit Kv channels [44], which could also lead to the limited vasodilatory effects of palmitic acid methyl ester in the SHRs, thus contributing to the PVAT-related reduction in vasorelaxation and the increasing vascular resistance and hypertension. In our previous experiment in the TA of HTG, we observed an antirelaxant effect of H_2_S produced by PVAT and the vascular wall, which contributed to endothelial dysfunction [21]. It seems that PVAT and PVAT-derived factors can be modulated by different pathological conditions, such as hypertriglyceridemia or hypertension, which are often associated with inflammation and increased levels of free radicals, contributing to endothelial dysfunction [21,45].

In our study, we showed that H_2_S produced by the same tissue (vascular wall) interfered with the regulation of different types of vasoactive (vasoconstriction vs. vasorelaxation) responses in opposite (pro-contractile vs. pro-relaxant effects). A cascade of reactions and interactions of H_2_S with distinct factors and enzymes leads to the initiation of vasorelaxation or vasoconstriction. The pro-contractile effects of H_2_S could be due to several signal pathways, such as the inhibition of the cAMP/adenylyl cyclase or the NO-synthase pathway, or the stimulation of a cyclooxygenase pathway. Similarly, the pro-relaxant effects of H_2_S could be associated with the activation of different mechanisms: hyperpolarization of different potassium channels, inhibition of endogenous phosphodiesterase, or stimulation of the NO-synthase pathway. In summary, endogenously produced H_2_S manifested a dual effect depending on the type of triggered signaling pathway, the detection of which will depend on further studies.

Several experimental works have demonstrated the antihypertensive properties of H_2_S through the application of an exogenous donor of H_2_S [19,22]. The compensatory effect of H_2_S was shown by the experiments of Cacanyiova et al. [10] and Berenyiova et al. [22] in the TA of SHRs. Compared to Wistar rats, SHRs had an increased vasorelaxant phase of the dual vasoactive responses to exogenously administered H_2_S, which was further potentiated by acute NO deficiency. However, it appears that the presence of PVAT may alter the vasoactive effects of exogenous H_2_S. In SHRs, unlike in Wistar rats, we confirmed that the presence of PVAT increased vasorelaxant responses of the TA, suggesting a close relationship between the vasorelaxant effect of H_2_S and PVAT. We observed similar results in the MA of SHRs, where the presence of PVAT significantly reduced the contractile part of Na_2_S-induced vasomotor responses and increased the vasorelaxant responses. One of the possible signaling pathways of H_2_S could be a specific group of KCNQ channels that are opened by the ADRF, which in turn leads to vasorelaxation. Kohn et al. [35] confirmed that they can also be activated by exogenous H_2_S. Thus, exogenous H_2_S donors could be a suitable compensatory therapeutic agent to open these channels in pathological conditions associated with ADRF disorder.

## 5. Conclusions

In this study, we demonstrated that PVAT can play an important role in the regulation of vascular tone under hypertensive conditions. We confirmed that, although PVAT of the TA in SHRs aggravated endothelial function, it revealed an anticontractile effect mediated by the release of unknown factors. Moreover, the presence of PVAT increased the vasorelaxant response to H_2_S donors. Endogenously produced H_2_S manifested a dual effect depending on the type of signaling pathway triggered. H_2_S produced by PVAT and the vascular wall had pro-contractile effects and could contribute to pathological changes in essential hypertension. However, H_2_S produced by the vascular wall had a pro-relaxation effect and could represent a form of vasoactive compensatory mechanism to balance impaired vascular tone regulation. The presence of compensatory vasoactive mechanisms is also confirmed by the finding that, although we observed increased oxidative stress within the cardiovascular system, this did not affect endothelial function.

## Figures and Tables

**Figure 1 biomolecules-12-00457-f001:**
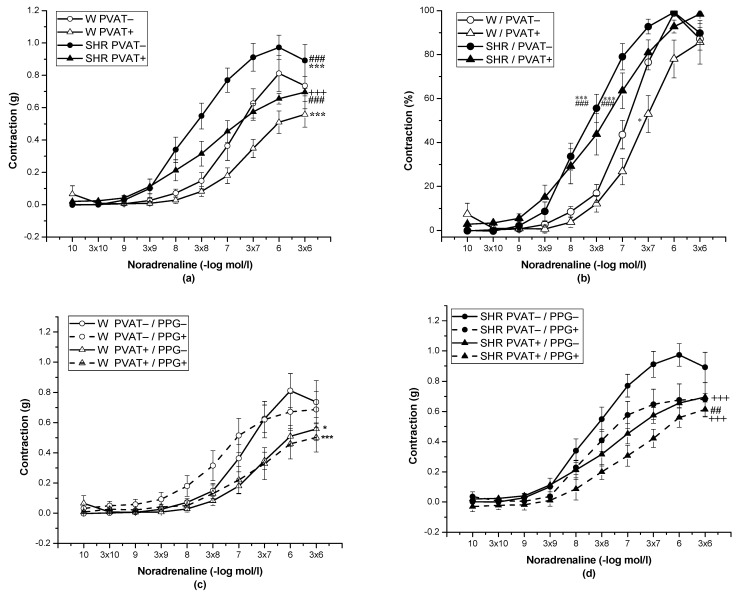
Nordrenaline-induced contractile response of the thoracic aorta in normotensive Wistar rats (W, *n* = 8) and spontaneously hypertensive rats (SHRs, *n* = 8) with preserved (PVAT+) or denuded (PVAT−) perivascular adipose tissue expressed as the active wall tension (**a**), and as percentages of the maximal-reached response induced by noradrenaline (**b**), before (PPG−) and after (PPG+) pretreatment with DL-propargylglycine (**c**,**d**). The results are presented as the mean ± S.E.M., and differences between groups were analyzed by three-way ANOVA with the Bonferroni post hoc test on ranks. *** *p* < 0.001 vs. W PVAT−; +++ *p* < 0.001 vs. SHR PVAT−; ### *p* <0.001 vs. W PVAT+; * *p* < 0.05 vs. W PVAT−; * *p* < 0.05 vs. W PVAT−/PPG−; *** *p* < 0.001 vs. W PVAT−/PPG−; +++ *p* < 0.001 vs. SHR PVAT−/PPG−; ## *p* < 0.01 vs. SHR PVAT+/PPG−.

**Figure 2 biomolecules-12-00457-f002:**
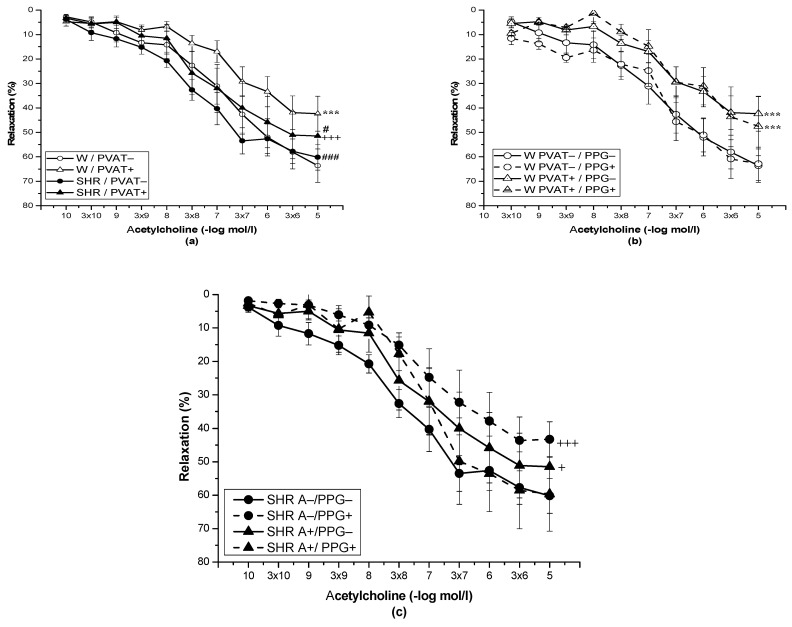
Endothelium-dependent vasorelaxation of the thoracic aorta of normotensive Wistar rats (W, *n* = 8) and spontaneously hypertensive rats (SHRs, *n* = 8) with intact (PVAT+) or removed (PVAT−) perivascular adipose tissue (**a**) before (PPG−) and after (PPG+) pretreatment with DL-propargylglycine (**b**,**c**). The results are presented as the mean ± S.E.M., and differences between groups were analyzed by three-way ANOVA with the Bonferroni post hoc test on ranks. *** *p* < 0.001 vs. W PVAT−/PPG−; +++ *p* < 0.001 vs. SHR PVAT−/PPG−; # *p* < 0.05 vs. W PVAT+; ### *p* < 0.001 vs. W PVAT+; + *p* < 0.05 vs. SHR PVAT−/PPG−.

**Figure 3 biomolecules-12-00457-f003:**
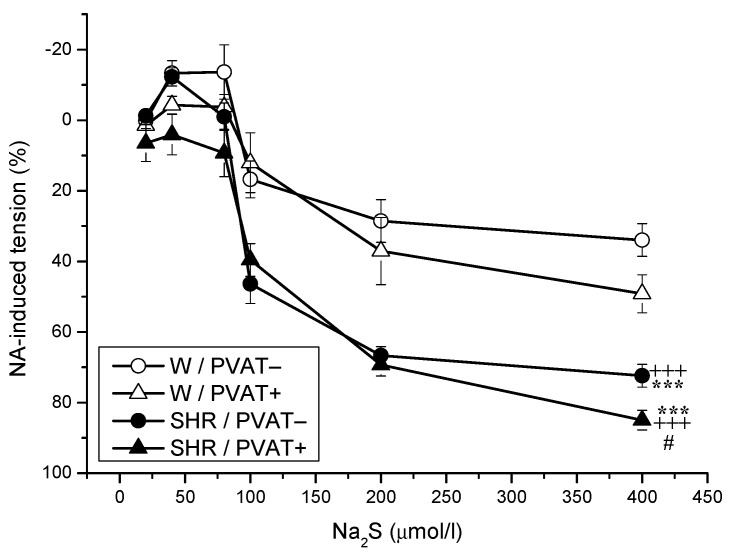
Dual vasoactive response of the thoracic aorta induced by sodium sulfide (Na_2_S), an exogenous donor of H_2_S in normotensive Wistar rats (W, *n* = 8) and spontaneously hypertensive rats (SHRs, *n* = 8) with intact (PVAT+) or removed (PVAT−). The results are presented as the mean ± S.E.M., and differences between groups were analyzed by three-way ANOVA with the Bonferroni post hoc test on ranks. # *p* < 0.05 vs. SHR PVAT−; *** *p* < 0.001 vs. W PVAT−; +++ *p* < 0.001 vs. W PVAT+.

**Figure 4 biomolecules-12-00457-f004:**
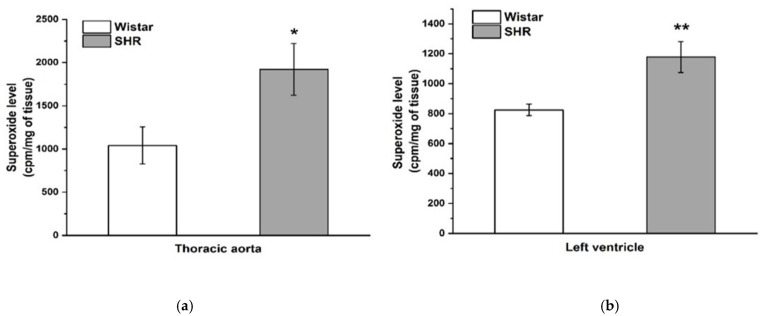
The production of superoxide in the thoracic aorta (**a**) and left ventricle (**b**) in Wistar and spontaneously hypertensive rats. The results are presented as the mean ± S.E.M., and differences between groups were analyzed by Student’s *t*-test. * *p* < 0.05, ** *p* < 0.01 vs. Wistar.

**Figure 5 biomolecules-12-00457-f005:**
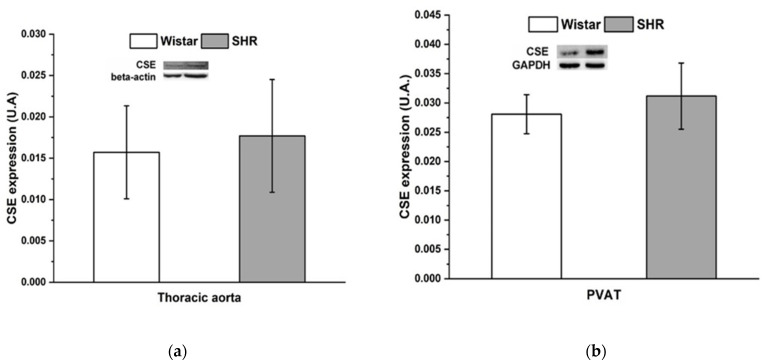
The level of cystathionine γ-lyase protein expression in the thoracic aorta (**a**) and perivascular adipose tissue in Wistar and spontaneously hypertensive rats (**b**). The results are presented as the mean ± S.E.M.

**Table 1 biomolecules-12-00457-t001:** General characteristics of experimental animals.

Parameter	Wistar Rats	Spontaneously Hypertensive Rats
SBP (mmHg)	128.14 ± 2.82	172.68 ± 2.22 **
BW (g)	424.5 ± 10.77	335.5 ± 0.73 ***
HW (mg)	1240 ± 28.81	1275 ± 26.47
TL (mm)	39.48 ± 0.51	37.16 ± 0.47
HW/BW (mg/g)	2.92 ± 0.05	3.80 ± 0.04 ***
HW/TL (mg/mm)	31.42 ± 0.66	34.17 ± 0.88 *
RFW (mg)	3320.38 ± 547.13	3060.25 ± 132.91
RFW/TL (mg/mm)	83.68 ± 12.98	81.29 ± 3.95
Chol (mmol/L)	2.05 ± 0.18	2.19 ± 0.10
HDL (mg/dL)	56.92 ± 5.44	64.98 ± 2.69
LDL (mg/dL)	10.33 ± 1.42	13.1 ± 0.56
TAG (mmol/L)	2.23 ± 0.13	1.39 ± 0.18 **
GLU (mmol/L)	7.48 ± 0.36	7.36 ± 0.41

SBP, systolic blood pressure; BW, body weight; HW, heart weight; TL, tibia length; HW/BW, heart weight/body weight ratio; HW/TL, heart weight/tibia length ratio; RFW, retroperitoneal fat weight; RFW/TL, retroperitoneal fat weight/tibia length ratio; Chol, total cholesterol; HDL, high-density lipoprotein cholesterol; LDL, low-density lipoprotein; TAG, triacylglycerols; GLU, glucose. Values are shown as the mean ± S.E.M. * *p* < 0.05; ** *p* < 0.01; *** *p* < 0.001 vs. Wistar rats.

**Table 2 biomolecules-12-00457-t002:** Sensitivity of the thoracic aorta of Wistar and spontaneously hypertensive rats to noradrenaline.

	**Wistar Rats**				**Spontaneously Hypertensive Rats**			
	PVAT−		PVAT+		PVAT−		PVAT+	
	PPG−	PPG+	PPG−	PPG+	PPG−	PPG+	PPG−	PPG+
EC_50_	7.12 ± 0.09	7.68 ± 0.08 **	6.83 ± 0.06	7.0 ± 0.18	7.81 ± 0.15 **	7.49 ± 0.31	7.63 ± 0.24 *	7.12 ± 0.17

EC_50_, the negative logarithm of NA molar concentration inducing the half-maximum response; NA, noradrenaline; PPG, DL-propargylglycine. Values are mean ± S.E.M. * *p* < 0.05 vs. Wistar PVAT− PPG−; ** *p* < 0.01 vs. Wistar PVAT− PPG−.

**Table 3 biomolecules-12-00457-t003:** The effect of acute pretreatment with propargylglycine (10 mmol/L) on basal tension of the thoracic aorta in Wistar and spontaneously hypertensive rats.

DL-propargylglycine (g)	**Wistar Rats**		**Spontaneously Hypertensive Rats**	
	PVAT−	PVAT+	PVAT−	PVAT+
	0 ± 0.00	−0.08 ± 0.02	0.14 ± 0.06 *	−0.02 ± 0.03

PVAT, perivascular adipose tissue. Positive values = increase in basal tension; negative values = decrease in basal tension. * *p* < 0.05 vs. Wistar PVAT−.

## Data Availability

All data arising from this study are contained within the article, and any additional data sharing will be considered by the first author upon request.

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
