# Peer review of "The Vasoactive Effect of Perivascular Adipose Tissue and Hydrogen Sulfide in Thoracic Aortas of Normotensive and Spontaneously Hypertensive Rats"

_biomolecules, 2022, doi:10.3390/biom12030457_

Round 1
Reviewer 1 Report
The authors studied on the role of parivascular adipose tissue on the regulation of vaso-activity (thoracic aorta) in comparison of of normotensive and spontaneously hypertensive rats. The work is quite interesting. However, there is one point should be amended.
the Legend of Fig. 5. the last words: "and differences between groups" what does it mean?. Please describe clearly.
Author Response
Thank you for revising our MS and for remarks.
The Legend of Fig. 5. the last words: "and differences between groups" what does it mean?. Please describe clearly.
We corrected it. It was a mistake, we deleted a part "and differences between groups".
Reviewer 2 Report
- The relevance of the manuscript is logically described in sufficient details in the introduction section. However, it is not disclosed why the production of the superoxide anion was additionally investigated and what is a relation between superoxide anion and H2S. This issue is little mentioned in the conclusion section. But it seems logical to indicate this also in the introduction section.
- Authors provide the measurement of blood lipids. However, the purpose of such measurement is not entirely clear from the text of the manuscript. Has any hypothesis been put forward about the relationship between the plasma lipoproteins/lipids level and hydrogen sulfide production or hydrogen sulfide action on the vessels? It seems that it would be better to clarify the purpose of these studies in the manuscript.
- The “Functional study” section describes the method for measuring the contractile activity of the vessels. Authors don’t indicate the method of endothelium removal and how they confirm successful removal of endothelium or normal activity of endothelium in endothelium-intact vessels.
- Na2S was used as a hydrogen sulfide donor, which releases H2S upon dissociation. It is known that the dissociation of Na2S in an aqueous solution results in the formation of NaOH, which alkalizes the medium. This may affect the resulting physiological effects (for example, Farrukh I.S et al. Effect of pH on pulmonary vascular tone, reactivity, and arachidonate metabolism, J Appl Physiol,. 1989 Jul;67(1):445-52 or Kim UC et al. Effects of pH on Vascular Tone in Rabbit Basilar Arteries, J Korean Med Sci. 2004 Feb; 19(1): 42–50). Did the authors monitor the Na2S influence on pH of the medium to exclude pH-dependent effects?
- In the “conclusion” section, line 483, the authors write "H2S produced by PVAT, and the vascular wall had pro-contractile effects", and in line 485 "H2S produced by the vascular wall had a pro-relaxation effect". Hence it is not clear whether the hydrogen sulfide produced by the vascular wall has a pro-relaxing or pro-contractile effect. It is worth clarifying the conditions leading to pro-contractile or pro-relaxing effects.
Author Response
Reviewer 2
Thank you for revising our MS and for remarks. The comments were accepted and the mistakes were corrected.
The relevance of the manuscript is logically described in sufficient details in the introduction section. However, it is not disclosed why the production of the superoxide anion was additionally investigated and what is a relation between superoxide anion and H2S. This issue is little mentioned in the conclusion section. But it seems logical to indicate this also in the introduction section.
We agree with the reviewer. We added a missing information to the introduction and conclusion. P.2 l. 79-83, P.3 l. 108-113, P.15 l. 523-525.
Authors provide the measurement of blood lipids. However, the purpose of such measurement is not entirely clear from the text of the manuscript. Has any hypothesis been put forward about the relationship between the plasma lipoproteins/lipids level and hydrogen sulfide production or hydrogen sulfide action on the vessels? It seems that it would be better to clarify the purpose of these studies in the manuscript.
We agree with the reviewer. We added a missing information to the introduction. P. 2 l. 60-70.
The “Functional study” section describes the method for measuring the contractile activity of the vessels. Authors don’t indicate the method of endothelium removal and how they confirm successful removal of endothelium or normal activity of endothelium in endothelium-intact vessels.
In our study we used endothelium-preserved arteries only. The presence of functional endothelium was assessed in all preparations by determining the ability of acetylcholine (10–5 mol/L) to induce relaxation in noradrenaline (NA) pre-contracted (10–6 mol/l) arteries. We added this information to the MS. P.4 l.157-159, 161.
Na2S was used as a hydrogen sulfide donor, which releases H2S upon dissociation. It is known that the dissociation of Na2S in an aqueous solution results in the formation of NaOH, which alkalizes the medium. This may affect the resulting physiological effects (for example, Farrukh I.S et al. Effect of pH on pulmonary vascular tone, reactivity, and arachidonate metabolism, J Appl Physiol,. 1989 Jul;67(1):445-52 or Kim UC et al. Effects of pH on Vascular Tone in Rabbit Basilar Arteries, J Korean Med Sci. 2004 Feb; 19(1): 42–50). Did the authors monitor the Na2S influence on pH of the medium to exclude pH-dependent effects?
We agree with the opponent that under certain circumstances H2S can affect the pH of the environment. In the past, we examined the relationship between H2S and pH, where we examined the effect of pH on H2S-induced responses (Ondrias et al. DOI 10.1007/s00424-008-0519-0). However, already during those experiments, we confirmed that the administration of H2S did not significantly change the pH of the incubation solution (pH was measured directly in the pneumoxid-oxygenated solution at 37°C).
In the “conclusion” section, line 483, the authors write "H2S produced by PVAT, and the vascular wall had pro-contractile effects", and in line 485 "H2S produced by the vascular wall had a pro-relaxation effect". Hence it is not clear whether the hydrogen sulfide produced by the vascular wall has a pro-relaxing or pro-contractile effect. It is worth clarifying the conditions leading to pro-contractile or pro-relaxing effects.
We added a missing explanation to the MS. P.14 l.485-496.